# Adversarial Attack for SAR Target Recognition Based on UNet-Generative Adversarial Network

**Chuan Du and Lei Zhang \***

School of Electronics and Communication Engineering, Sun Yat-sen University, Shenzhen 518107, China; duchuan@mail.sysu.edu.cn
\* Correspondence: zhanglei57@mail.sysu.edu.cn

**Abstract:** Some recent articles have revealed that synthetic aperture radar automatic target recognition (SAR-ATR) models based on deep learning are vulnerable to the attacks of adversarial examples and cause security problems. The adversarial attack can make a deep convolutional neural network (CNN)-based SAR-ATR system output the intended wrong label predictions by adding small adversarial perturbations to the SAR images. The existing optimization-based adversarial attack methods generate adversarial examples by minimizing the mean-squared reconstruction error, causing smooth target edge and blurry weak scattering centers in SAR images. In this paper, we build a UNet-generative adversarial network (GAN) to refine the generation of the SAR-ATR models' adversarial examples. The UNet learns the separable features of the targets and generates the adversarial examples of SAR images. The GAN makes the generated adversarial examples approximate to real SAR images (with sharp target edge and explicit weak scattering centers) and improves the generation efficiency. We carry out abundant experiments using the proposed adversarial attack algorithm to fool the SAR-ATR models based on several advanced CNNs, which are trained on the measured SAR images of the ground vehicle targets. The quantitative and qualitative results demonstrate the high-quality adversarial example generation and excellent attack effectiveness and efficiency improvement.

**Keywords:** adversarial attack; adversarial example generation; UNet; generative adversarial network (GAN); synthetic aperture radar (SAR); automatic target recognition (ATR)

## 1. Introduction

As an active imaging sensor, synthetic aperture radar (SAR) has the advantages of collecting all-time, all-weather, high-resolution images [1–3]. SAR-automatic target recognition (ATR) is a vital method to extract remote sensing information and plays an essential role in earth monitoring, military and homeland security [4–7]. In the field of SAR-ATR, deep convolutional neural networks (CNNs) have been proven powerful tools due to their hierarchical feature extraction ability [8–12]. However, several works have revealed that some security problems exist in these SAR-ATR models.

Szegedy et al. [13] first discover that by injecting well-designed tiny perturbations into image samples, adversarial examples can be intentionally produced to cause the recognition model to misclassify. This process of generating adversarial examples is named as "adversarial attack", which has become a recent study trend [14–19] in the research field of remote sensing, radar, radio, etc. In radar signal processing, [14,15] verify that high-resolution range profile (HRRP) and SAR image target recognition models can be attacked successfully by well-designed adversarial examples. A faster C&W adversarial attack algorithm [16] is proposed to effectively fool deep CNN-based SAR target classifiers and meet real-time requirements. In the field of remote sensing, Li et al. [17] provide abundant experiments and insightful analysis on the adversarial attack of the deep CNNs-based remote sensing image scene classification. The work [18] systematically analyzes the

influence of adversarial examples on classification results of remote sensing scene classifiers based on deep neural networks (DNNs), which also demonstrates that the defense capability of the classifiers to the adversarial examples can be significantly improved by adversarial training. In terms of radio propagation, white-box and black-box adversarial attack methods are explored in [20], showing the vulnerability of radio signals classification based on DNNs to adversarial examples. Due to the openness of wireless communication, the end-to-end learning communication system based on auto-encoders can be easily destroyed by the well-designed adversarial perturbations [21]. Although several adversarial attack algorithms have been proposed to generate adversarial examples, generating them with high efficiency requires more exploration.

Various adversarial attack algorithms have been proposed in recent years. For example, as a gradient-based method, the fast gradient sign method (FGSM) [22] produces adversarial examples by taking a one-step update of the original image along with the sign of the gradient of the cross-entropy classification loss function. The basic iterative method (BIM) [23] and projected gradient descent (PGD) [24] are the iterative versions of FGSM, which utilize the multiple steps gradient information to obtain better attack effectiveness. The DeepFool [25] finds the closest distance from the input image to the target classification boundary and performs an iterative attack to perturb the original image beyond the classification boundary. However, the defensive distillation algorithm [26] can defense against these existing adversarial attacks except the C&W attack [27]. As an optimization-based method, the C&W attack [27] models the adversarial examples generation as an optimization process that maximizing the confidence of the adversarial examples labeled as a wrong category while minimizing the power of the adversarial perturbations (mean-squared reconstruction error (MSE) loss). The C&W has acquired excellent adversarial attack performance. According to the attributed scattering center model, a SAR image of a target can be regard as the sum of the responses from various individual scattering centers in different range-Doppler cells [28]. Hence, the C&W's MSE loss function is not suitable for SAR image adversarial example generation tasks, which will cause smooth target edge and blurry weak scattering centers in SAR image adversarial examples. Moreover, it is not appropriate for the adversarial attack task requiring an instant response, since its iterative optimization process costs a lot of time.

To efficiently generate adversarial examples of SAR images with sharp target edges and explicit weak scattering centers, in this paper, we propose to train a generator and discriminator in an adversarial way. We build a UNet [29] to realize the generator, which can extract the separable features of the targets from the whole SAR images to influence the recognition results. Moreover, it concatenates the low-resolution and high-resolution feature maps and learns the basic component scattering center information to generate a more refined SAR image adversarial examples. The discriminator aims to encourage that the generated adversarial examples are approximate to the real SAR images in sense of data distribution. In general, we apply the generative adversarial networks (GANs) [30] to efficiently produce high-quality adversarial examples for SAR images in white-box condition by adversarial training.

Our contributions are listed as the following.

(1) We leverage a generator to generate adversarial examples through fast network mapping rather than the iterative optimization in the previous optimization-based methods. Therefore, the proposed adversarial attack algorithm provides the SAR-ATR system with real-time attack capability.
(2) We utilize the UNet to learn the separable features of the targets to cause the misclassification of the recognition model. The UNet can also fuse the multi-resolution feature maps, benefiting the generation of SAR image adversarial examples.
(3) By introducing a discriminator, we can train the generator to produce higher-quality adversarial examples for SAR images by adversarial training, which can possess sharper target edges and more explicit weak scattering centers and achieve better attack performance.

The rest parts of this article are arranged as follows. Section 2 describes the problem definition of adversarial attack and our proposed algorithm in detail. In Section 3, we evaluate our proposed models and report experimental results. Conclusions and future works waiting to be explored are in Section 4.

## 2. Preliminaries

*Adversarial Attack for SAR-ATR*

Supposing $\mathcal{X}$ is the SAR image dataset. $x_n \in \mathbb{R}^{W \times H}$ is the $n$-th SAR image sample and $y_n$ is the corresponding ground truth category label of $x_n$ in the dataset $\mathcal{X}$, where $W$ and $H$ denote the width and height of the SAR image, respectively. $F(\cdot)$ is a target recognition model that provide a correct category prediction of a SAR image. For a commonly used deep CNN recognition model $F(\cdot)$ with a softmax output layer, given an input SAR image sample $x$, the output of $F(x)$ is $p \in \mathbb{R}^S$ denoting the probability distribution of the predicted categories, where $p_s \in [0,1]$, $\sum_{s=1}^{S} p_s = 1$ and $S$ denotes the number of the total target categories. The index of the predicted target category is an integer $C(x) = \arg\max_s (F(x)_s) \in [1, 2, ..., S]$.

The aim of an adversarial attack for SAR-ATR is to generate the corresponding adversarial example $\tilde{x}$ and make the SAR-ATR model misclassify. Meanwhile, $\tilde{x}$ needs to be approximate to the original SAR image $x$ under some distance metric so that their differences would not be perceived easily, where $\tilde{x} = x + \delta$, and $\delta$ is the added tiny adversarial perturbation. The whole frameworks of SAR-ATR and adversarial attack for SAR-ATR are shown in Figure 1.

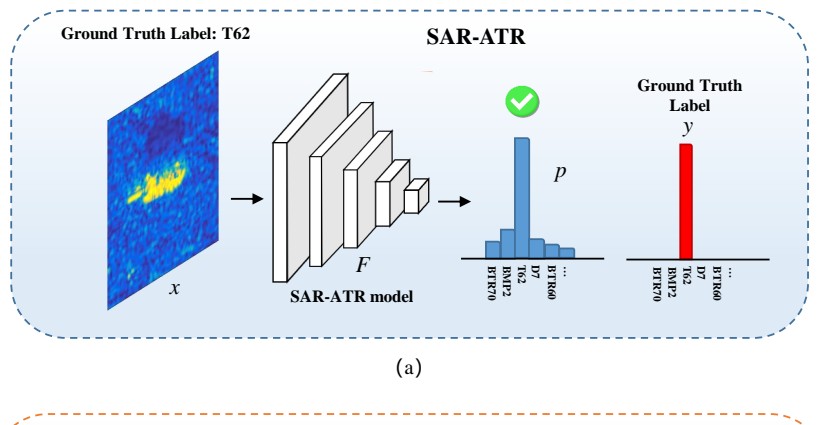

(a)

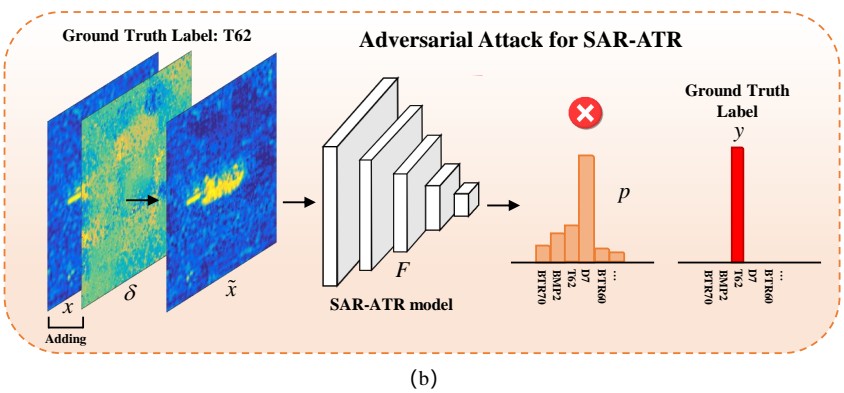

(b)

**Figure 1.** The whole framework of the SAR-ATR and the adversarial attack for SAR-ATR. (**a**) SAR-ATR, (**b**) adversarial attack for SAR-ATR.

The commonly-used adversarial attack modes are introduced below.

**Targeted attack:** If there is a SAR image $x$ and a designated category $t \neq y$, targeted attack aims to find an adversarial example $\tilde{x}$ which is similar to $x$, subject to $C(\tilde{x}) = t$. Namely, the targeted attack can cause the SAR-ATR model to mislabel the adversarial example as the designated category.

**Non-targeted attack:** If there is no designated category for the adversarial example, the adversarial attack is reduced to a search for the adversarial example $\tilde{x}$ which is similar to the original SAR image $x$, subject to $C(\tilde{x}) \neq y$, which is called a non-targeted attack.

## 3. Method

According to the attributed scattering center model, a SAR image of a target can be regard as the sum of the responses from various individual scattering centers in different range-Doppler cells [28]. The existing adversarial attack methods in SAR-ATR obtain the adversarial examples by minimizing the mean reconstruction square error, which will lead to the smooth target edge and the blurry weak scattering centers in the generated SAR image. These generated adversarial examples are obviously different from the real SAR images and possess poor deception. Therefore, to produce the adversarial examples with the characteristics of SAR images (with sharp target edges and explicit weak scattering centers), we propose an Attack-UNet-GAN algorithm to improve the quality of the generated adversarial example.

### 3.1. Attack-UNet-GAN

The overall architecture of our proposed Attack-UNet-GAN algorithm is shown in Figure 2b, which is consisted of three modules: a generator $G(\cdot)$, a discriminator $D(\cdot)$ and a SAR-ATR model $F(\cdot)$. In the optimization-based adversarial attack algorithms, the adversarial examples of the test SAR images are generated by re-optimizing the loss function iteratively, which is of high time cost. Therefore, to construct the fast mapping from the original SAR images to the adversarial examples, we build the generator $G(\cdot)$. The input of $G(\cdot)$ is the original SAR image $x$ and its output is the adversarial example $\tilde{x} = G(x)$. $G(\cdot)$ aims to learn the basic component scattering center information in the SAR images and encourage the generated adversarial examples to be indistinguishable from the original SAR images for the discriminator $D(\cdot)$. Furthermore, to generate more realistic adversarial examples with the characteristics of SAR image, the generated adversarial example $\tilde{x}$ is then sent into the discriminator $D(\cdot)$, whose function is to distinguish the generated adversarial example $\tilde{x}$ from the real SAR images $x$ as possible. Thus, $G(\cdot)$ and $D(\cdot)$ are trained in an adversarial way and strengthen each other. To achieve the task of attacking the SAR-ART model, we should first have a high-accuracy SAR-ART model $F(\cdot)$, which takes the adversarial example $\tilde{x}$ as input and outputs its predicted target label.

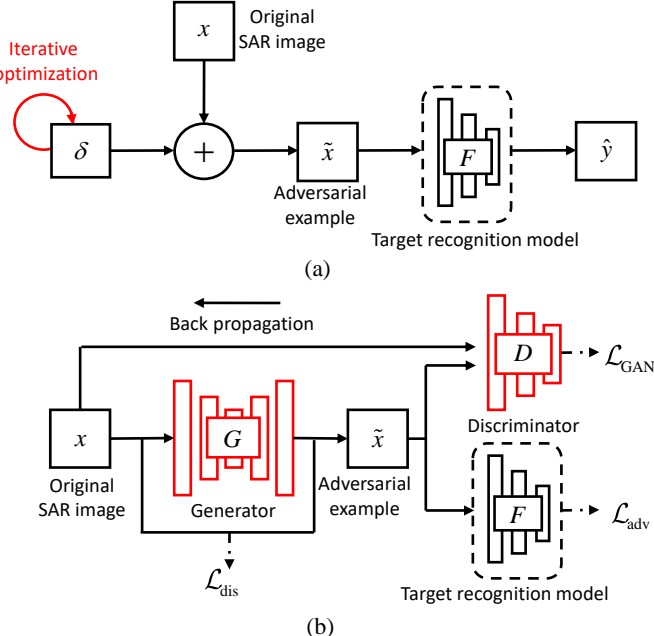

**Figure 2.** The comparison of the algorithms' frameworks: (**a**) C&W and (**b**) Attack-UNet-GAN.

3.1.1. The Design of Loss Function

**Targeted Attack:** According to Section 2, we can describe the process of a targeted attack as the following constrained optimization problem:

$$\min_{\tilde{x}} \ \mathcal{L}_{\text{dis}}(\tilde{x}, x), \ \text{s.t.} \ C(\tilde{x}) = t, \ \tilde{x} \in [0, 1]^{W \times H} \tag{1}$$

where $\mathcal{L}_{\text{dis}}$ is the distance metric used for the similarity measurement between the original SAR image and the adversarial example. In this way, the differences between $x$ and $\tilde{x}$ are limited to be as small as possible. Due to the non-differentiable characteristic of the constraint $C(\tilde{x}) = t$, (1) is difficult to be solved, so some mathematical transformations are needed. Here, we construct a function $g_1(\cdot)$ to make $C(\tilde{x}) = t$ and $g_1(\tilde{x}) \leq 0$ equivalent [27]. In this paper, the following expression of $g_1(\cdot)$ is used:

$$g_1(\tilde{x}) = \left( \max_{s \neq t}(F(\tilde{x})_s) - F(\tilde{x})_t \right)^+, \tag{2}$$

where $s$ denotes the integer index of the target category. $F(\tilde{x})_t$ denotes the predicted classification probability of the designated category $t$. The $(z)^+$ is a shortened form of $\max(z, 0)$. Thus, we replace (1) by the following:

$$\min_{\tilde{x}} \ \mathcal{L}_{\text{dis}}(\tilde{x}, x), \ \text{s.t.} \ g_1(\tilde{x}) \leq 0, \ \tilde{x} \in [0, 1]^{W \times H} . \tag{3}$$

Then, the method of Lagrange multipliers is applied, and we can derive the loss function as follows:

$$\begin{aligned} \mathcal{L} &= \min_{\tilde{x}}(\mathcal{L}_{\text{dis}} + \mathcal{L}_{\text{adv}}) \\ &= \min_{\tilde{x}}(\mathcal{L}_{\text{dis}}(\tilde{x}, x) + \lambda \cdot g_1(\tilde{x})) \end{aligned} . \tag{4}$$

For the box constraint $\tilde{x} \in [0, 1]^{W \times H}$ in (3), we use the sigmoid activation function to solve it, which is described in detail in Section 3.1.3. In this paper, we utilize the $l_2$-norm: $\|v\|_2 = \sqrt{\sum_{i=1}^{n} \left( |v_i|^2 \right)}$ to realize $\mathcal{L}_{\text{dis}}$ in (4). Then, the optimization problem waited to be solved is

$$\mathcal{L} = \min_{\tilde{x}}(\|\tilde{x} - x\|_2 + \lambda \cdot g_1(\tilde{x})), \tag{5}$$

where $g_1(\tilde{x})$ describes the distance between $\max_{s \neq t}(F(\tilde{x})_s)$ and $F(\tilde{x})_t$. Minimizing the value of $g_1(\tilde{x})$ will encourage the perturbed SAR image to be mislabeled as the designated category $t$. Minimizing the value of $\|\tilde{x} - x\|_2$ aims to limit the power of the added adversarial perturbation. Note that the adversarial attack loss $\mathcal{L}_{\text{adv}}$ should describe the opposite of the difference between the predicted probability of the designated category $t$ and the highest predicted probability among the other categories in the targeted attack, or the difference between the predicted probability of the ground truth category $y$ and the highest predicted probability among the other categories in the non-targeted attack [27].

A smaller $g_1(\tilde{x})$ means the probability of the input SAR image classified as the designated category $t$ is higher. If $\max_{s \neq t}(F(\tilde{x})_s)$ is optimized to be just smaller than $F(\tilde{x})_t$, the optimization of the loss function (5) will be finished. Hence, we introduce a threshold $\eta$ to form a soft hinge loss on $\mathcal{L}_{\text{adv}}$ as

$$g_1(\tilde{x}) = \max \left( -\eta, \max_{s \neq t}(F(\tilde{x})_s) - F(\tilde{x})_t \right), \tag{6}$$

which means that the optimization will not be stopped until $\max\limits_{s \neq t}(F(\tilde{x})_s)$ is $\eta$ smaller than $F(\tilde{x})_t$. It can elevate the final probability of the input SAR image classified as the designated category $t$.

In this way, the loss function in targeted attack mode can be represented as:

$$\mathcal{L} = \min_{\tilde{x}} \left( \|\tilde{x} - x\|_2 + \lambda \cdot \max\left( -\eta, \max_{s \neq t}(F(\tilde{x})_s) - F(\tilde{x})_t \right) \right). \tag{7}$$

**Non-targeted Attack:** For a non-targeted attack, there is no designated category for the adversarial examples to be mislabeled as. In this way, the loss function in the non-targeted attack just aims to realize the misclassification of an adversarial example to a wrong category. Therefore, we should construct the $\mathcal{L}_{\text{adv}}$ in the loss function to describe the difference between the predicted probability of the ground truth category $y$ and the highest predicted probability among the other categories [27]. Here, we replace (4) by the following loss function in the non-targeted attack:

$$\begin{aligned} \mathcal{L} &= \min_{\tilde{x}}(\mathcal{L}_{dis} + \lambda \cdot \mathcal{L}_{adv}) \\ &= \min_{\tilde{x}}(\|\tilde{x} - x\|_2 + \lambda \cdot g_2(\tilde{x})) \end{aligned} \tag{8}$$

where $g_2(\tilde{x})$ measures the difference between the predicted probability of the wrong category and the predicted probability of the ground truth category $y$. The selected expression of $g_2(\tilde{x})$ is as the following:

$$g_2(\tilde{x}) = \left( F(\tilde{x})_y - \max_{s \neq y}(F(\tilde{x})_s) \right)^+, \tag{9}$$

where $F(\tilde{x})_y$ denotes the probability of the adversarial example $\tilde{x}$ recognized as the ground truth category $y$. In order to elevate the robustness of the algorithm and the misclassification confidences of the adversarial examples, we also introduce a threshold $\eta$. Then $g_2(\tilde{x})$ can be written as:

$$g_2(\tilde{x}) = \max\left( -\eta, F(\tilde{x})_y - \max_{s \neq y}(F(\tilde{x})_s) \right), \tag{10}$$

which means that the optimization of adversarial example $\tilde{x}$ will not be stopped, until $F(\tilde{x})_y$ is $\eta$ smaller than $\max\limits_{s \neq y}(F(\tilde{x})_s)$.

Hence, the loss function for optimizing the adversarial example $\tilde{x}$ in the non-targeted attack can be written as:

$$\mathcal{L} = \min_{\tilde{x}} \left( \|\tilde{x} - x\|_2 + \lambda \cdot \max\left( -\eta, F(\tilde{x})_y - \max_{s \neq y}(F(\tilde{x})_s) \right) \right). \tag{11}$$

### 3.1.2. The Introduction of UNet and GAN

To realize the real-time generation of the adversarial example, a generator $G_\theta(\cdot)$ is built as shown in Figure 2, replacing the iterative searching process of the adversarial example in the C&W algorithm. The input of the generator $G_\theta(\cdot)$ is the original SAR image $x$, and its output is the adversarial example $\tilde{x} = G_\theta(x)$. Therefore, the $\tilde{x}$ in (7) and (11) can be replaced by $G_\theta(\tilde{x})$. Moreover, to generate the adversarial examples with the characteristics of SAR images, we also introduce a discriminator $D_\phi(\cdot)$ to construct an adversarial training method, as shown in Figure 2. The input of the discriminator is the adversarial example $\tilde{x}$ or original SAR image $x$, and its output is a scalar indicating whether the input is a real SAR image. The generator aims to generate the adversarial example $\tilde{x}$ that is similar to the real SAR image $x$ in the sense of data distribution, so as to deceive the discriminator $D_\phi(\cdot)$.

The goal of the discriminator $D_\phi(\cdot)$ is to distinguish the adversarial example $\tilde{x}$ from the original SAR image $x$. So the adversarial training loss [30] can be expressed as:

$$\begin{aligned}
\mathcal{L}_{\text{GAN}} &= \mathbb{E}_x \log D_\phi(x) + \mathbb{E}_x \log\big(1 - D_\phi(\tilde{x})\big) \\
&= \mathbb{E}_x \log D_\phi(x) + \mathbb{E}_x \log\big(1 - D_\phi(G_\theta(x))\big)
\end{aligned} \tag{12}$$

where we denote the $\mathbb{E}_x \equiv \mathbb{E}_{x \sim p_{data}}$, $\theta$ and $\phi$ are the parameters of the generative network and discriminative network, respectively. Note that the adversarial training loss aims to encourage the data distribution of the generated adversarial examples approximate to that of the real SAR images, leading to the generated adversarial examples with sharp target edges and explicit weak scattering centers. Thus, the generated adversarial examples possess the characteristics of SAR images and are highly deceptive.

Finally, the whole loss function of the algorithm Attack-UNet-GAN can be expressed as the following:

$$\mathcal{L} = \mathcal{L}_{\text{dis}} + \lambda \cdot \mathcal{L}_{\text{adv}} + \mathcal{L}_{\text{GAN}}, \tag{13}$$

where $\lambda > 0$ is a suitably chosen constant that controls the relative importance of misclassification loss $\mathcal{L}_{\text{adv}}$.

For the optimization of the networks $G_\theta(\cdot)$ and $D_\phi(\cdot)$, we solve the min-max game $\arg\min_\theta\max_\phi\mathcal{L}$ by alternating the iterative optimizations of $G_\theta(\cdot)$ and $D_\phi(\cdot)$. When we fix the parameters $\phi$ and optimize $\min_\theta\mathcal{L} = \min_\theta(\mathcal{L}_{\text{dis}} + \lambda \cdot \mathcal{L}_{\text{adv}}) + \min_\theta(\mathcal{L}_{\text{GAN}})$, the optimization of the first term $\min_\theta(\mathcal{L}_{\text{dis}} + \lambda \cdot \mathcal{L}_{\text{adv}})$ can make the target recognition model misclassify and the added adversarial perturbations imperceptible. The optimization of the second term $\min_\theta(\mathcal{L}_{GAN}) = \min_\theta\big(\mathbb{E}_x \log\big(1 - D_\phi(G_\theta(x))\big)\big)$ can make the output of $D_\phi(G_\theta(x))$ approach 1, that is, $G_\theta(\cdot)$ can make the generated adversarial examples $G_\theta(x)$ similar to the real SAR images. When we fix the parameters $\theta$ and optimize $\max_\phi\mathcal{L} = \max_\phi(\mathcal{L}_{GAN}) = \max_\phi\big(\mathbb{E}_x \log D_\phi(x) + \mathbb{E}_x \log\big(1 - D_\phi(G_\theta(x))\big)\big)$, the optimization of it can make the output of $D_\phi(x)$ approach 1 and the output of $D_\phi(G_\theta(x))$ approach 0, that is, $D_\phi(\cdot)$ can make the discriminator distinguish the real SAR images and the generated adversarial examples as possible. Thus, the generator and discriminator can be optimized in an adversarial way. The whole training process is outlined in Algorithm 1. Once $G_\theta(\cdot)$ is well-trained on the training SAR image data and SAR-ATR model, it can produce an adversarial example for each test SAR image through the one-step network mapping instead of the iterative optimization of the adversarial perturbations.

---

**Algorithm 1** An example for the training process in Attack-UNet-GAN algorithm

---

1: Train a high-accuracy deep CNN-based SAR-ATR model $F(\cdot)$.
2: Build a generator $G_\theta(\cdot)$ and a discriminator $D_\phi(\cdot)$.
3: Initialize generator parameters $\theta$ and discriminator parameters $\phi$.
4: Set the appropriate mini-batch size, learning rate and network parameters and so on.
5: **for** number of training iterations **do**
6:     Sample a mini-batch of $m$ SAR images $\left\{x^{(1)}, ..., x^{(m)}\right\}$ from the training dataset, whose corresponding ground truth labels are $\left\{y^{(1)}, ..., y^{(m)}\right\}$;
7:     Feed the mini-batch of SAR images into the generator $G_\theta(\cdot)$ to generate the adversarial examples $\left\{\tilde{x}^{(1)}, ..., \tilde{x}^{(m)}\right\}$ and get the loss function $\mathcal{L}_{\text{dis}}$;
8:     Feed the generated adversarial examples into the SAR-ATR model $F(\cdot)$ to output the prediction probability for each category and get the loss function $\mathcal{L}_{\text{adv}}$;
9:     Feed the generated adversarial examples and original SAR images into the discriminator $D_\phi(\cdot)$ alternately and get the loss function $\mathcal{L}_{\text{GAN}}$;
10:     Update the parameters $\theta$ and $\phi$ by SGD alternately on the whole loss function $\arg\min_\theta\max_\phi\mathcal{L}$ in (13)
11: **end for**

---

### 3.1.3. The Detailed Network Architecture

**The discriminator *D*:** The detailed network architecture of the discriminator *D* is shown in Figure 3. Its input is the adversarial example $\tilde{x}$ or original SAR image *x* with the size of $128 \times 128$. There are five blocks in the discriminator, each of which contains a $4 \times 4$ convolutional layer with the stride size of 2 and padding size of 1 followed by a batch-normalization layer. The activation function between each block is a leaky ReLU function. The output layer is a sigmoid layer and outputs a scalar indicating whether the input sample is an original SAR image.

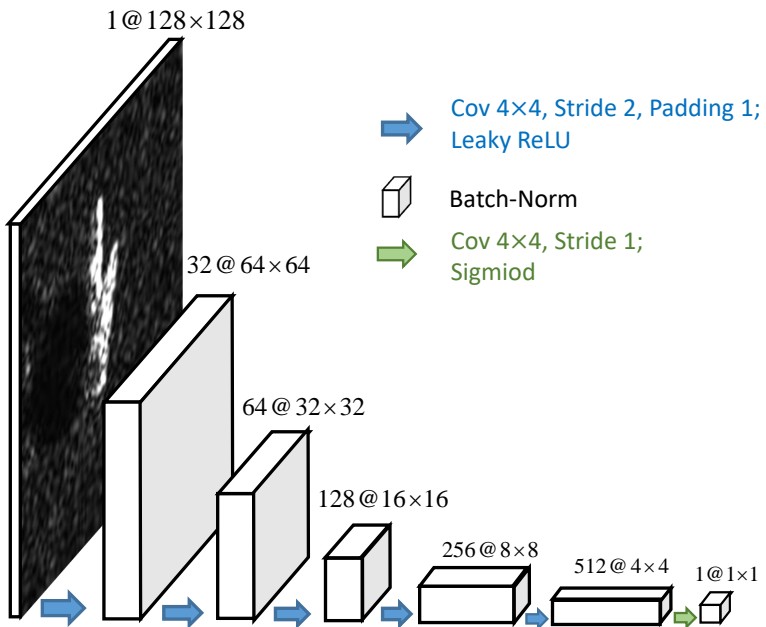

**Figure 3.** The network architecture of the discriminator *D* in detail.

**The generator *G*:** In this paper, we introduce a UNet to realize the generator *G* and generate the adversarial examples. The UNet is initially proposed for the task of medical image segmentation [29]. It is a symmetric U-shaped deep CNN, which is consisted of an encoder and a decoder. The encoder is a stack of convolution, activation and pooling layers learning the separable features and basic component scattering center information. The sizes of the extracted feature maps are reduced by the encoder, which are then progressively expanded by the decoder. The decoder realizes the SAR image adversarial example generation with transposed convolutions. It combines the different resolution feature maps by a sequence of up-convolutions and the concatenation with corresponding feature maps of the same size from the encoder. This combination causes more sufficient feature information to be propagated to the higher resolution layers of the decoder, which can benefit the precise SAR image adversarial example generation.

The detailed UNet architecture is illustrated in Figure 4. The network architecture is symmetric for the encoder and decoder. The size of the input SAR image is $128 \times 128$. The number of layers, the resolution of each feature map and the number of feature map channels are also shown in Figure 4. The block of the encoder is consisted of two $3 \times 3$ convolution layers followed by a $2 \times 2$ max-pooling layer. The block of the decoder contains a $2 \times 2$ up-convolution layer followed by two $3 \times 3$ convolution layers. The output is the adversarial example, whose size is equal to that of the input SAR image. Note that we take the sigmoid layer as the output layer of the UNet, to restrict the values of the generated adversarial examples to the range of [0, 1].

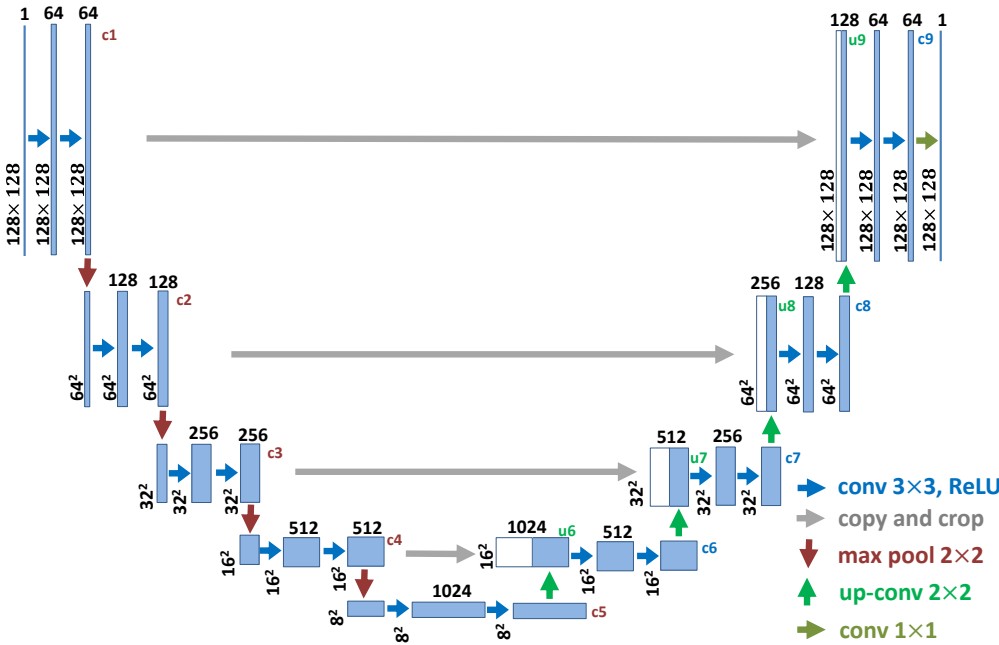

**Figure 4.** The network architecture of the generator *G* (UNet) in detail.

## 4. Experiment

In this section, we use the well-trained SAR-ATR models on the public measured SAR image data to verify and test our proposed adversarial attack algorithm. We compare its attack performance with others competitive adversarial attack algorithms by attacking these deep CNN models. The experiments prove our algorithm's competitive effectiveness, excellent efficiency and high-quality adversarial example generation.

### 4.1. Dataset and Experimental Setup

4.1.1. Dataset

The famous public measured SAR image data of the ground vehicle targets, the moving and stationary target acquisition and recognition (MSTAR) dataset [31,32], is utilized in our experiment. It is provided by the Air Force Research Laboratory and the Defence Advanced Research Projects Agency (AFRL/DARPA) [31]. This SAR image dataset is acquired leveraging the X-band HH polarization "STARLOS" spotlight SAR platform with the resolution of 0.3 m × 0.3 m. As the significant dataset for SAR-ATR performance evaluation, it contains abundant SAR images of vehicle targets and ground clutter. There are ten categories of vehicle targets in the dataset, such as BTR70, BTR60, BRDM2 and BMP2 (armored personnel carrier); 2S1 (rocket launcher); D7 (bulldozer); ZIL131 (truck); T62 and T72(tank); ZSU234 (air defense unit) [33], which are indexed by category labels 1, 2, ..., 10, respectively. These SAR images in each category cover all target-aspect angles in the range of [0°, 360°] with a relative flat grass or exposed soil background. The adjacent target-aspect angle intervals are within [1°, 2°]. Notice that all targets are stationary targets. The optical images and corresponding SAR images of the targets are displayed in Figure 5.

We rescale the collected SAR images as 128 × 128 pixels and obtain 5950 slice images. Each slice image is labeled as one of the ten kinds of targets. In addition, we carry out the amplitude normalization pre-processing to guarantee that the value of each SAR image pixel is limited within the range of [0, 1]. To validate the proposed algorithm's generalization capability, the target-depression angles of the training and test SAR images are different. The target-depression angles and the numbers of the training and test images before the data augmentation are also listed in Table 1. In the training phase of the SAR-ATR models, the commonly used training data augmentation techniques [10], such as pose synthesis, translation and speckle noising, are also applied to alleviate the effects of overfitting and get the high-accuracy SAR-ATR models. Specifically, we first use one SAR image

to produce 10 synthesized pose SAR images (rotating the SAR images). Then, they are translated by five times randomly. Finally, we perform the speckle noising augmentation operations on each translated SAR image with the parameter *a* (the maximum intensity of noise samples) set as 0.5, 1.0 and 1.5 [10].

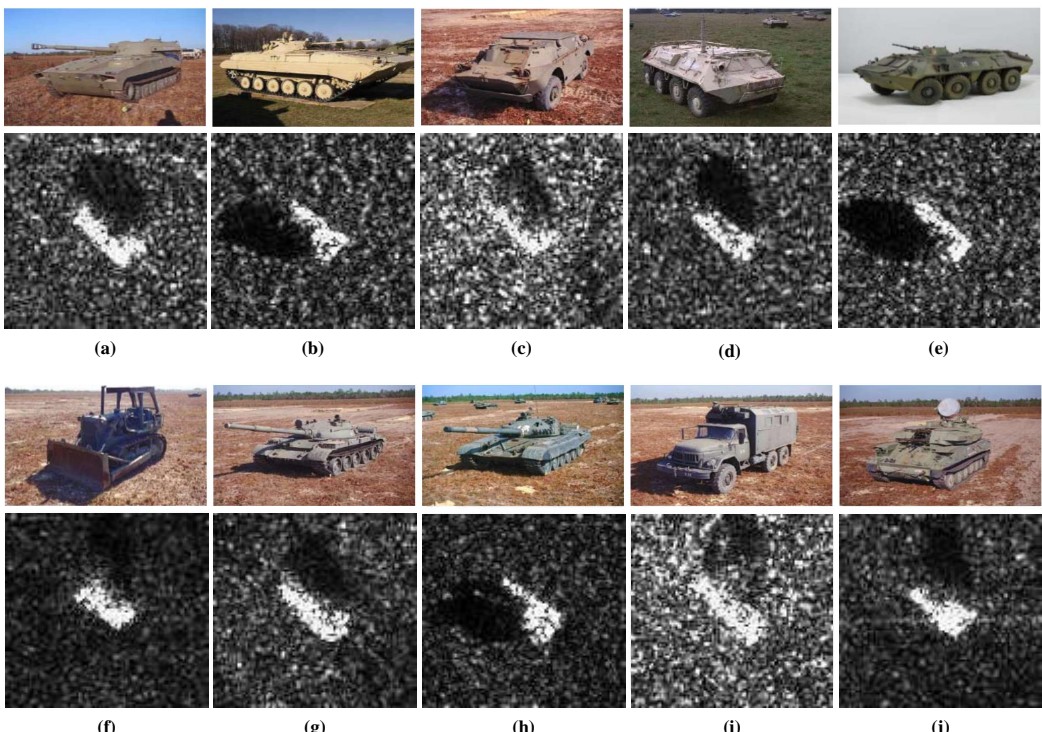

**Figure 5.** SAR images of all the ground targets in the MSTAR dataset and their corresponding optical images. (**a**) 2S1, (**b**) BMP2, (**c**) BRDM2, (**d**) BTR60, (**e**) BTR70, (**f**) D7, (**g**) T62, (**h**) T72, (**i**) ZIL131, (**j**) ZSU234.

**Table 1.** The Numbers and target-depression angles of the training and test SAR images for SAR-ATR before the data augmentation.

| Class | Training | | Testing | |
|---|---|---|---|---|
| | **Depression Angles** | **Number** | **Depression Angles** | **Number** |
| 2S1 | 15° | 196 | 17° | 233 |
| BRDM2 | 15° | 196 | 17° | 233 |
| BTR60 | 15° | 196 | 17° | 232 |
| D7 | 15° | 195 | 17° | 256 |
| T72 | 15° | 274 | 17° | 299 |
| BMP2 | 15° | 274 | 17° | 298 |
| BTR70 | 15° | 274 | 17° | 299 |
| T62 | 15° | 273 | 17° | 299 |
| ZIL131 | 15° | 274 | 17° | 299 |
| ZSU234 | 15° | 274 | 17° | 299 |

4.1.2. Baselines and Experimental Setup

The following adversarial attack algorithms are the baselines compared with our algorithm:

- **Fast Gradient Sign Method (FGSM) [22]:** Adversarial examples are generated by taking one-step update of the input along with the sign of the gradient of the cross-entropy loss function.

- **Basic Iterative Method (BIM) [23]:** It is an extension of the FGSM by running a finer optimization for multiple iterations.
- **Project Gradient Descent (PGD) [24]:** It is an iterative version of the FGSM, which takes multiple small steps iteratively while randomly adjusts the updating direction after each step.
- **DeepFool [25]:** It finds the closest distance from the original image to the classification boundary and performs an iterative attack to perturb the original image beyond the classification boundary.
- **Carlini and Wagner's Attack (C&W) [27]:** The adversarial examples are generated by maximizing the probability of the adversarial example labeled as a wrong category while minimizing the power of the adversarial perturbations.

For the attacked SAR-ATR model, we use the standard deep learning classifiers, AlexNet [34], VGGNet16 [35] and ResNet32 [36], which are trained on the MSTAR dataset and have a classification accuracy of over 96%. The generator $G$ is realized by a UNet [29] making the output and input SAR image size the same, whose detailed architecture is shown in Figure 4. For the discriminator $D$, the deep CNN [37] shown in Figure 3 is utilized to achieve it. For the distance metric function in this paper, we choose the $l_2$-norm. To optimize the generator and discriminator parameters, we adopt the Adam optimizer [38] with the learning rate $10^{-4}$, the hyperparameter $\beta_1 = 0.5$, $\beta_2 = 0.999$, and the training batch size 64. We carry out all experiments in a Python program on a personal computer with a 3.7 GHz CPU, a 64 GB RAM, and a 24 GB NVIDIA Geforce RTX 3090 GPU.

*4.2. Evaluation Measurements*

Suppose that there are $N$ test SAR images that can be classified correctly by the SAR-ATR model in total. The adversarial examples are generated from these $N$ SAR images in the test dataset.

**Targeted Attack:** The attack success rate in the targeted attack mode is calculated by the following formula:

$$\text{Acc}_{\text{targeted}} = \sum_{n=1}^{N} \text{I}\left(C(\tilde{\mathbf{x}}_n) = t^{(n)}\right) \Big/ N, \tag{14}$$

where $\text{I}(\cdot)$ denotes the indication function, $C(\tilde{\mathbf{x}}_n)$ is the predicted category of the adversarial example $\tilde{\mathbf{x}}_n$, and $t^{(n)}$ is the designated category of the $n$-th adversarial example.

**Non-targeted Attack:** Without the designated category, the attack success rate in the non-targeted attack mode is calculated by the following formula:

$$\text{Acc}_{\text{non-targeted}} = \sum_{n=1}^{N} \text{I}\left(C(\tilde{\mathbf{x}}_n) \neq y^{(n)}\right) \Big/ N, \tag{15}$$

where $y^{(n)}$ is the ground truth category of the $n$-th original SAR image.

*4.3. Attack Performance Comparison*

In this experiment, we attack different SAR-ATR models based on the deep CNNs (AlexNet, VGGNet16, ResNet32) under the condition of the white-box attack, which means that the network structures and parameters of the recognition models are known. The attack success rates of different adversarial attack algorithms for different recognition models in targeted and non-targeted attack modes are shown in Tables 2 and 3, respectively. The attack success rates can reflect the effectiveness of the adversarial attack algorithms.

Among these adversarial attack algorithms, FGSM, BIM, PGD and DeepFool are gradient-based algorithms. C&W and Attack-GAN belong to optimization-based algorithms. BIM has higher attack success rates than FGSM, because it utilizes the multiple-step gradient information to acquire a more precise optimization result. PGD performs better than BIM, since it not only takes multiple small steps gradient update iteratively as BIM, but also randomly adjusts the direction after each step to search for a better adversarial example. The attack success rates of Attack-UNet-GAN are much higher than those of

FGSM and competitive with those of the other four baseline algorithms. Attack-UNet-GAN can attack the SAR-ATR models more successfully than the Attack-UNet (without discriminator *D*). Due to the introduction of the discriminator *D*, the adversarial training loss can improve the data description ability of the generator *G* to generate better adversarial examples. In terms of attack success rate, Attack-UNet performs better than Attack-CNN, whose generator is realized by an 8-layer CNN with $1 \times 1$ convolution kernels, since the UNet fuses the multiple resolution feature maps' information and helps the more sufficient feature information be propagated to the higher resolution layers of the decoder to generate the better adversarial examples.

**Table 2.** The performances of different adversarial attack algorithms for different SAR-ATR models in terms of attack success rate under the condition of targeted attack.

| Targeted Attack | AlexNet | VGGNet16 | ResNet32 |
|---|---|---|---|
| FGSM | 95.34 | 87.19 | 75.55 |
| BIM | 98.08 | 97.41 | 97.84 |
| PGD | **98.73** | **98.01** | 98.57 |
| DeepFool | 97.91 | 97.02 | 98.26 |
| C&W | 98.59 | 97.84 | **98.62** |
| Attack-CNN | 94.79 | 93.86 | 94.38 |
| Attack-UNet | 97.85 | 97.26 | 98.11 |
| Attack-Unet-GAN | **98.47** | **97.63** | **98.39** |

**Table 3.** The performances of different adversarial attack algorithms for different SAR-ATR models in attack success rate under the condition of non-targeted attack.

| Non-Targeted Attack | AlexNet | VGGNet16 | ResNet32 |
|---|---|---|---|
| FGSM | 96.21 | 88.34 | 77.62 |
| BIM | 98.56 | 97.53 | 98.02 |
| PGD | **98.91** | **98.15** | 98.52 |
| DeepFool | 98.31 | 97.25 | 98.17 |
| C&W | 98.77 | 97.92 | **98.69** |
| Attack-CNN | 95.03 | 94.45 | 94.91 |
| Attack-UNet | 97.93 | 97.14 | 98.02 |
| Attack-UNet-GAN | **98.59** | **97.74** | **98.57** |

### 4.4. Comparison of the Generation Speed

To compare the calculation efficiency of each adversarial attack algorithm, we generate adversarial examples of the same test SAR image with $128 \times 128$ pixels under the same calculation condition and record the running time of each algorithm's program. The time cost of generating a $128 \times 128$ pixels SAR image's adversarial example for different adversarial attack algorithms is shown in Table 4. Among all these algorithms, the algorithms based on our proposed framework possess the fastest adversarial example generation speed, since they gain the adversarial example through the fast network mapping of the generator, rather than the iterative optimization in the C&W algorithm or the multiple calculations of the input test SAR images' gradients in the BIM or PGD algorithm. Especially, compared with the C&W algorithm, the generation speed of the adversarial example for Attack-UNet-GAN is improved hundreds of times.

**Table 4.** Time complexity of different algorithms for generating an adversarial example for a $128 \times 128$ pixels SAR image.

| | FGSM | BIM | PGD | C&W | DeepFool | Attack-CNN | Attack-UNet | Attack-UNet-GAN |
|---|---|---|---|---|---|---|---|---|
| **Time (s)** | 0.0091 | 0.6152 | 0.4985 | 0.8537 | 0.2863 | **0.0025** | **0.0039** | **0.0039** |

### 4.5. Influence of the Constant $\lambda$

To study the influence of the constant $\lambda$ in (13) on the attack performance, we use the Attack-UNet-GAN algorithm to attack the SAR-ATR model based on ResNet32 for the values of $\lambda$ located uniformly (on the log scale) from $\lambda = 0.001$ to $\lambda = 100$. We plot the attack success rates and MSE distances for different values of $\lambda$ in Figure 6. We can see that when $\lambda \leq 0.01$, the attack rarely succeeds. The attack success rate gradually increases to almost 100%, when the value of $\lambda$ varies from 0.01 to 1. When $\lambda \geq 1$, the differences between the original SAR images and the generated adversarial examples become more apparent, but the attack always succeeds. Therefore, in our experiments, we set the value of $\lambda$ as 1 to weigh the deception and attack performance of the generated adversarial examples.

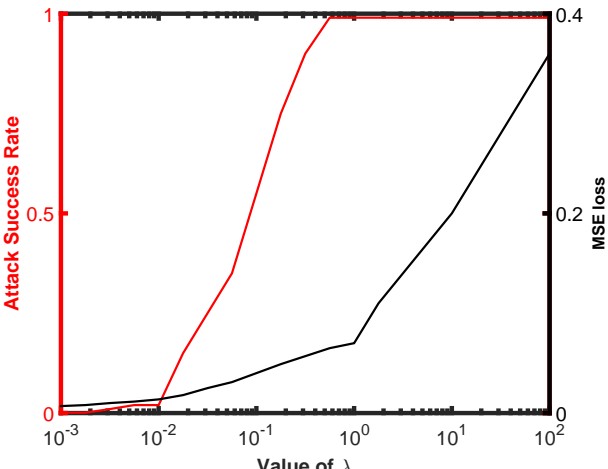

**Figure 6.** The influence of the constant $\lambda$ on the attack performance. We plot the attack success rate and MSE loss between the adversarial example and original SAR images as a function of $\lambda$.

### 4.6. Visualization of the Adversarial Examples

In this section, we carry out experiments to show the deception performace of the generated adversarial examples by different attack algorithms. The generated adversarial examples and the corresponding adversarial perturbations by different adversarial attack algorithms in targeted and non-targeted attacks are shown in Figures 7 and 8. The attacked SAR-ATR model is based on the same ResNet32 for all adversarial attack algorithms. The predicted categories of the adversarial examples by the high-accuracy SAR-ATR model and the misclassified confidences to the wrong category for different adversarial attack algorithms are shown above the corresponding adversarial examples. We can see that the adversarial perturbations of FGSM, PGD, and BIM cover most parts of the SAR images. For C&W and DeepFool, the adversarial perturbations are mainly located on the SAR images' shadow regions. Attack-UNet and Attack-UNet-GAN can mainly concentrate the adversarial perturbations on the target regions of the SAR images, because the target region of a SAR image possesses much more separable information benefiting the target recognition task than the background clutter and shadow regions. Thus, Attack-UNet and Attack-UNet-GAN can learn and utilize this separable information through the generator $G$ to help produce the adversarial examples and fool the SAR-ATR model. In Figures 7g and 8g, the target edges are sharper and the weak scattering centers of the target are more explicit than those in Figures 7h and 8h, such as the regions surrounded by the red ellipses. Because the introduction of the discriminator $D$ can help the generated adversarial examples approximate to real SAR images in the sense of data distribution and make them possess the characteristics of SAR images. For the DeepFool attack, there are some generated adversarial examples that can fool the SAR-ATR model successfully. However, the added adversarial perturbations are too strong, making the differences between the original SAR images and adversarial examples conspicuous.

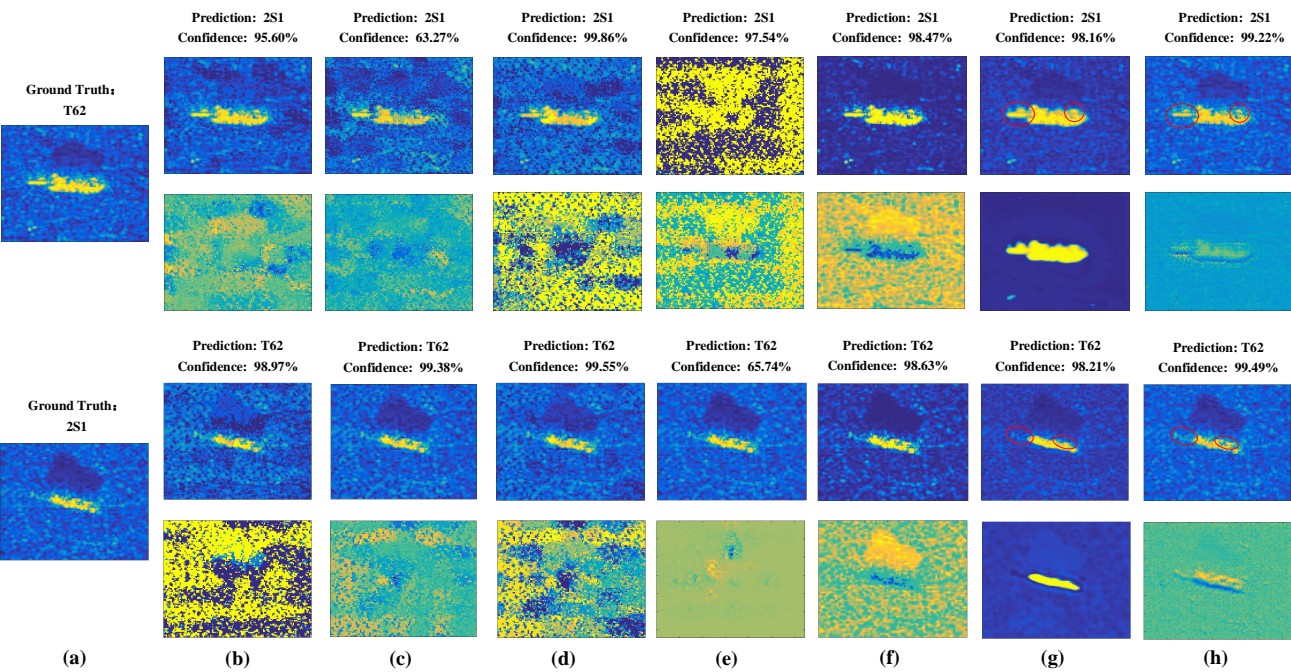

**Figure 7.** (**a**) Original SAR images of the targets. The first row shows the adversarial examples produced by different attack algorithms in the targeted attack mode; the second row shows the corresponding adversarial perturbations. The predicted categories and confidences of the adversarial examples are listed above them. The corresponding attack algorithms: (**b**) FGSM, (**c**) BIM, (**d**) PGD, (**e**) DeepFool, (**f**) C&W, (**g**) Attack-UNet and (**h**) Attack-UNet-GAN.

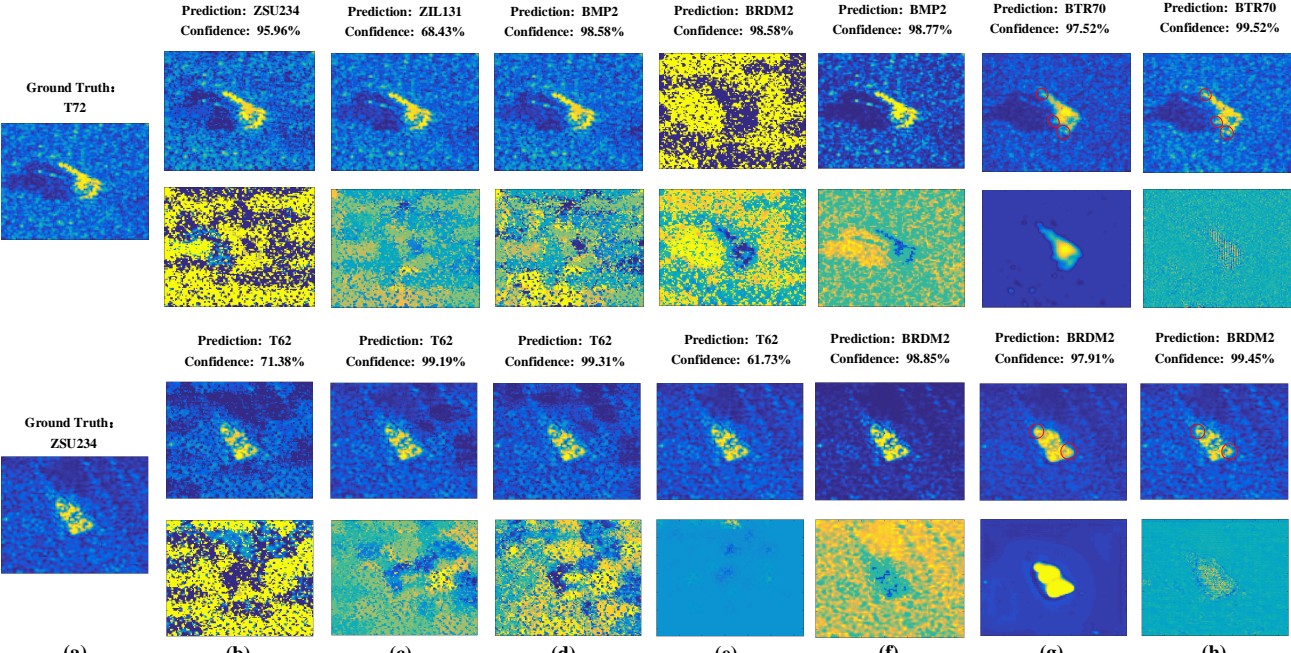

**Figure 8.** (**a**) Original SAR images of the targets. The first row shows the adversarial examples produced by different attack algorithms in the non-targeted attack mode; the second row shows the corresponding adversarial perturbations. The predicted categories and confidences of the adversarial examples are listed above them. The corresponding attack algorithms: (**b**) FGSM, (**c**) BIM, (**d**) PGD, (**e**) DeepFool, (**f**) C&W, (**g**) Attack-UNet and (**h**) Attack-UNet-GAN.

### 4.7. Display of the Learned Features in UNet

To exhibit the excellent target feature extracting ability of the UNet for the SAR images, we visualize the hierarchical representations of the SAR image features extracted

by different CNN layers of the UNet in Figure 9. In the first row of Figure 9, they are the features from the UNet's encoder. It can be observed that the closer the layer is to the input original SAR image, the more specialized the leaned features are. On the contrary, the farther the layer is to the input SAR image, the more fundamental the leaned features are. The features of the fourth layer (Figure 9(c4)) in the encoder can be regarded as different basic strong scatter centers to construct all of the SAR target images. The features of the third layer (Figure 9(c3)) in the encoder are the component structures used to constitute the SAR images of the targets, such as spheres, dihedrals, trihedral, corner diffractions, etc. Further, the features learnt by the first layer (Figure 9(c1)) in the encoder possess more structure information, we can find different regions of the SAR image, such as the target, shadow and clutter regions. In the second row of Figure 9, they are the features from the UNet's decoder. It can be seen that the closer the layer is to the output adversarial example, the more specialized the learned feature are, which is symmetric to that of the UNet's encoder.

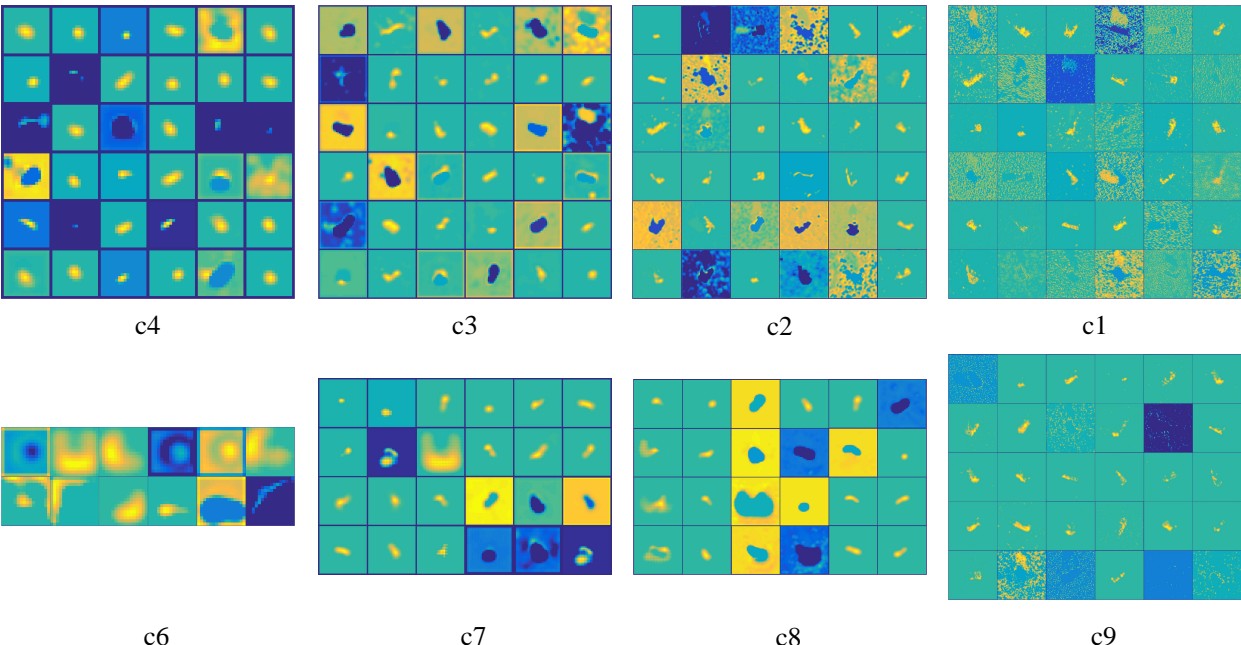

**Figure 9.** The display of the feature maps learned by the generator (UNet) in different layers. The first row displays the feature maps learned by the encoder of UNet; the second row displays the feature maps learned by the decoder of UNet. The corresponding layer indexes of the feature maps in Figure 3 are displayed below each feature map.

### 4.8. Separability of the Extracted Features

In this section, we represent the separability of the features extracted by the UNet. We visualize the original SAR images and high dimensional features extracted by the generator of Attack-UNet-GAN by utilizing T-SNE [39] to map them to the two-dimensional subspace in Figure 10a,b, respectively. The features are extracted by the last layer in the UNet's encoder (the layer c5 in Figure 4). In Figure 10, each dot represents a SAR image or the feature of a SAR image, and each color denotes a category. It can be seen that the features learned by the generator are more separable and discriminative than the original SAR images of the targets. That is, the generator of our model extracts the features with prominent separability, which can help generate adversarial examples and cause the SAR-ATR model to misclassify.

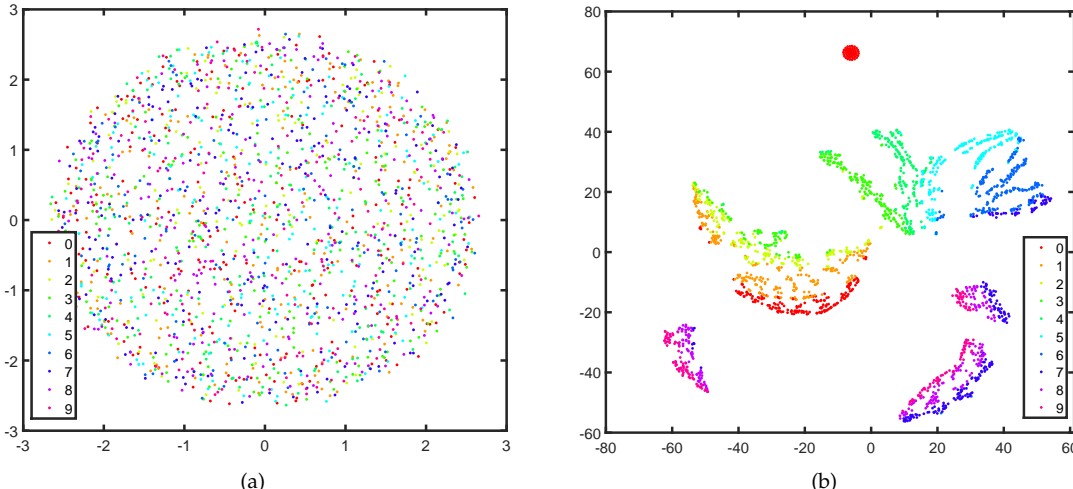

(a)  (b)

**Figure 10.** The separability of the original SAR images and features extracted by the generator (UNet) of Attack-UNet-GAN. The SAR images and the features are mapped to the two-dimensional subspace by T-SNE shown as the dots on the plane. Each dot represents a SAR image or the feature of a SAR image, and each color denotes a category. (**a**) the original SAR images, (**b**) the features extracted by the UNet.

### 4.9. Misclassified Category Distributions of the Adversarial Attack

To explore the misclassified category distribution of all adversarial examples, we calculate the misclassified categories for different adversarial attack algorithms. The misclassified category distributions show the percentages of the adversarial examples mislabelled as each of others target categories to all the adversarial examples of the ground truth label. We find that the misclassified categories are highly concentrated. As shown in Figures 11 and 12, we use pie charts to visualize the distributions of the misclassified categories of the adversarial attack algorithms based on the MSTAR SAR image dataset.

The misclassified category distributions of adversarial attack algorithms are shown in Figures 11 and 12. The ground truth category of all adversarial examples is D7 (bulldozer). Figure 11 shows the misclassified category distributions of the adversarial examples generated by six different attack algorithms for the same SAR-ATR model based on ReNet32. Figure 12 shows the misclassified category distributions of the adversarial examples generated by the same attack algorithm for three different deep CNN-based recognition models. In the pie charts, a color denotes a misclassified category. The percentage denotes the ratio of the number of the adversarial examples misclassified as the corresponding category to the total number of adversarial examples. In Figures 11 and 12, it can be seen that although the adversarial examples are generated by different adversarial attack algorithms or for different attacked recognition models, their major misclassified categories are almost the same. For example, the BRDM2 (armored personnel carrier) is the major misclassified category of the adversarial examples for the original SAR images of the D7 (bulldozer). The reasons for this phenomenon may be the homogeneity and heterogeneity among categories. As it is found in the work [40] that the misclassified categories of the adversarial examples are more probably to be the categories that are closer to them in the sample's feature space. Meanwhile, it can be observed that the similarity among the SAR images from different categories can be well reflected by the misclassified category distributions. For example, the armored personnel carrier is the major misclassified category of the adversarial examples from the bulldozer, representing that the armored personnel carrier and bulldozer may possess a strong similarity in the feature space or original SAR image space.



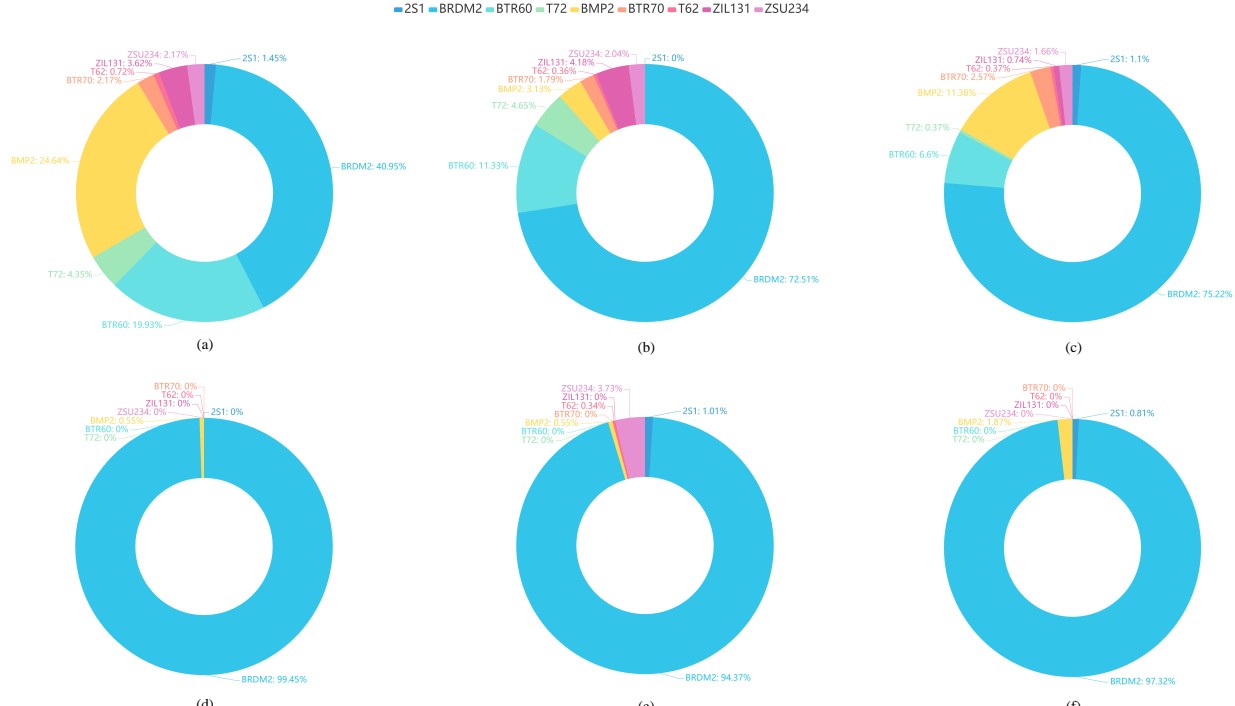

**Figure 11.** The misclassified category distributions of the adversarial examples (ground truth category: D7) generated by different attack algorithms for the same ResNet32-based SAR-ATR model. The misclassified category distributions show the percentages of the adversarial examples mislabelled as each of others target categories to all the adversarial examples of D7. (**a**) FGSM, (**b**) BIM, (**c**) PGD, (**d**) C&W, (**e**) DeepFool, (**f**) Attack-UNet-GAN.

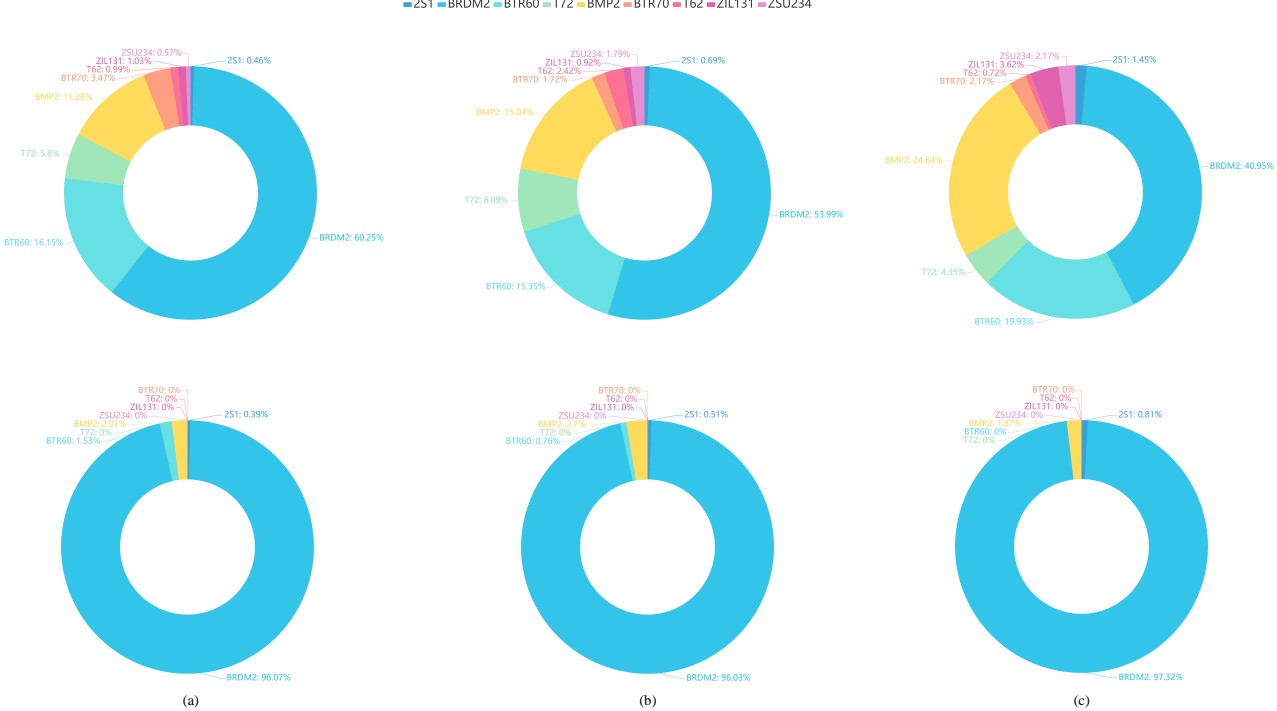

**Figure 12.** The misclassified category distributions of the adversarial examples (ground truth category: D7) generated by the same attack algorithms for different SAR-ATR models. The misclassified category distributions show the percentages of the adversarial examples mislabelled as each of others target categories to all the adversarial examples of D7. The first row shows the distribution of FGSM; the second shows the distribution of Attack-UNet-GAN. (**a**) AlexNet, (**b**) VGGNet16, (**c**) ResNet32.

## 5. Discussion

The experiment results of Section 4.4 demonstrate the excellent adversarial example generation speed. Compared with the C&W algorithm, the generation speed is promoted even hundreds of times. The reason is the utilization of the generative network's fast mapping. By utilizing a large number of training SAR images to train the generative network, it can well learn the basic features existing in SAR images to help build the mapping from the SAR image space to the adversarial example space. From the experiment results of Section 4.6, we observe that the introduction of GAN makes the generated SAR image adversarial examples possess sharp target edges and explicit weak scattering centers, because the adversarial training forces the generated adversarial examples to approximate the original SAR images in the sense of data distribution. Thus, the generated adversarial examples can possess the characteristics of real SAR images and stong deception. The experiment results of Sections 4.7 and 4.8 illustrate the UNet's powerful extraction capabilities of separable features and basic component scattering center information, which can benefit the generation of adversarial examples and cause the SAR-ATR model to misclassify.

From the experiment results of Section 4.3, we can conclude that the Attack-UNet-GAN algorithm has a competitive performance in terms of attack success rate with the baseline algorithms, since the baseline algorithms can update the adversarial examples iteratively leveraging the test data information. However, the Attack-UNet-GAN algorithm utilizes the well-trained generative network to yield the adversarial example in real-time, which is suitable for the adversarial attack of the SAR-ATR systems requiring instant responses. Therefore, we can study the improvement of the attack algorithm's generalization capability to make the algorithm has a higher attack success rate on the different test SAR images in the future. Moreover, the proposed algorithm can be improved further to provide help for jamming remote sensing monitoring system and deflecting important information acquisition from remote sensing images.

We also evaluate the attack performance on the measured SAR images dataset, Open-SARShip. We build the dataset by the SAR images of the ship targets, such as Cargo, Fishing, Tanker, Tug and Other-type. The numbers of the SAR images of Cargo, Fishing, Tanker, Tug and Other-type targets are 8130, 126, 1618, 172, 942, respectively. We use the half of each target's SAR images to construct the training dataset and the other half to construct the test dataset. The used SAR-ATR model is ResNet32. The average target classification accuracy is 78.48%. Then we use the baseline attack algorithms and Attack-UNet-GAN to attack the SAR-ATR model. We find that the generated adversarial examples are obviously different from the original SAR image of the target. Moreover, the attack success rates of these attack algorithm are very low. These attack algorithms do not perform well on the OpenSARShip dataset, probably because the resolutions of these SAR images are too low and the detailed information of the targets is not obvious. The adversarial attack algorithm can not make the SAR-ATR model misclassify by only modifying the original SAR image a little. That is, these attack algorithms are more suitable for attacking the SAR-ATR models of the high-resolution SAR images.

## 6. Conclusions

In this paper, an adversarial attack method based on UNet and GAN for deep learning SAR-ATR models is proposed. For our Attack-UNet-GAN algorithm, once well trained, the generator can produce adversarial examples efficiently through the network mapping for the test SAR images, replacing the time-consuming iterative re-optimization. By introducing the discriminator, the generated adversarial examples possess the characteristics of SAR images and are more deceptive, with sharper target edges and more explicit weak scattering centers. Utilizing the measured SAR image dataset, we demonstrate the strong attack performance of our algorithm in attack success rate, computation efficiency based on different deep learning recognition models. There are some potential future works to be explored. In practical applications, the relevant information of the SAR-ATR model is usually unknown, so it is more practical to propose a black-box adversarial attack al-

gorithm. We consider using the learning ability of the distillation network to construct such a black-box adversarial attack model. Moreover, the transferability of the generated adversarial examples for SAR images needs to be deeply explored. It is expected to propose an attack algorithm to generate the adversarial examples with strong transferability to attack more types of SAR-ATR models successfully.

**Author Contributions:** Conceptualization, C.D. and L.Z.; methodology, software, validation, writing—original draft preparation, writing—review and editing, visualization, data curation, C.D.; supervision, L.Z.; funding acquisition, C.D. and L.Z. All authors have read and agreed to the published version of the manuscript.

**Funding:** This work was supported in part by the National Nature Sciences Foundation of China under Grant 61771372, in part by the Shenzhen Science and Technology Program under Grant KQTD20190929172704911, and in part by the Open Fund of Science and Technology on Electromagnetic Scattering Key Laboratory under Grant 61424090112.

**Institutional Review Board Statement:** The study does not involve humans or animals.

**Informed Consent Statement:** The study does not involve humans.

**Data Availability Statement:** The experiment in this paper uses a public data set, so no data is reported in this work.

**Acknowledgments:** 

**Conflicts of Interest:** The authors declare that they have no conflict of interest to report regarding the present study.

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
