# Peer review of "Adversarial Attack for SAR Target Recognition Based on UNet-Generative Adversarial Network"

_remotesensing, doi:10.3390/rs13214358_

Round 1

Reviewer 1 Report

Authors tackle the comments pointed by R2 in their reviewed manuscript. In particular, they bring

  • better description of the MSTAR data
  • better description of the preprocessing
  • better description of the "adversarial attack framework"

However, they do not tackle neither R1 or R3 comments about adding experiments in an other dataset (e.g. opensarship).

Currently, if the framework works only with high resolution SAR image (which is not the case of opensarship), it is acceptable but it has to pointed.

For this reason, I support acceptance after minor revision (e.g. about testing the code on another dataset).

Author Response

Response: Please see the attachment.

Reviewer 2 Report

Authors redesigned the paper taking into account all the comments and suggestions. Now the manuscript is designed properly, results are clearly described, and the paper is ready for publication.  

Author Response

Thank you very much for your patience and work to this revised manuscript. We have improved the manuscript further.

Reviewer 3 Report

Thanks a lot for the revision. I think it has improved a lot. 

I would still suggest you to improve on the readability of figure 10, 11, and 12 specially on the text size and contrast.

Author Response

Response: Please see the attachment.

Reviewer 4 Report

In the paper authors proposed an adversarial attach method based on UNet and GAN for deep learning SAR-ATR models. Results and comparison with other methods have been provided in the paper indicating benefits of using the proposed method. 

Paper is interesting, well written and illustrated with results and comparisons between the proposed method and methods from literature. Except few minor corrections related to the English language, authors should also explain a bit better differences between Attack-CNN, attack-UNet and attack-UNet-GAN mentioned in Tables 2 and 3 and the reason why they compared them. In connection with this, in Figs.7 and 8 there are Attack-UNet and Attack-GAN and in Figs.11 and 12 there is only Attack-UNet-GAN.

Author Response

Response: Please see the attachment.

This manuscript is a resubmission of an earlier submission. The following is a list of the peer review reports and author responses from that submission.

Round 1

Reviewer 1 Report

The paper offers a GAN based adversarial attack targetting a SAR image classifier. The attack is compared to classical adversarial attack (FSGM, C&W...). However the offered attack has 2 important features

  • first, it is much faster (as the adversarial is produced directly by a generator and not by brute optimization)
  • and more importantly the attack is expected to produce a "SAR-like" image because the generator is trained (using GAN framework) to produce realistic image. This feature can not be guarantee by brute optimization of the image.

This last point seems crucial for physical image like SAR images. Indeed, in remote sensing, modifying the image can only be done by modifying the physical word. Thus, brute optimization may produce "impossible" images, and, may over estimate the real risk of adversarial attack. In this context, producing "realistic" adversarial seems more relevant, while, it seems not very common in litterature.

For these reasons, I think that the paper should to be published.

Now, the main weakness is the fact that all experiments are done on a single dataset. Evaluate some of the attack on openSAR-ship could greatly strenghen the paper.

Finally, for information, there is today much more stronger defense than distillation like for example https://arxiv.org/abs/1711.00851 which can more or less handle any norm based attack (but sometimes at the cost of a less efficient classifier on clean data).

Independently do you plan to put the code on a github page ?

Reviewer 2 Report

The manuscript presents interesting research on the use of deep neural network approaches for conducting the targeted and non-targeted attacks, by generation of adversarial SAR images. Although the study looks interesting and may contain experimental or methodological value in the areas of computer science, it does not fit into the topics of remote sensing. This article remains more technical report and it uses terms and description style, that more in line with the topic of computer vision or cyber security. To be published in the remote sensing journal, authors should make additional research and fully reshape the manuscript. I suggest a few comments about how it can be rewritten:

  1. Pay more attention on remote sensing details. Information about the data used in the research is very essential for the remote sensing topic. Without good data description and analysis in terms of remote sensing, this article can be published in the remote sensing journal.
  2. Section 2 is difficult to understand. Try to give first easy understandable definitions and description and only after this provide descriptions in terms of optimization theory and computer vision. Right now, this section is not understandable for the remote sensing community.
  3. Please describe in more details and justification sections on of loss functions for the generator and discriminator.
  4. From this paper it is difficult to understand the differences between target and non-target attacks. Please, try to describe it in more general way with examples.
  5. Please place section with dataset description before methodology. It is difficult to understand the ideas of provided algorithm and how this methodology works, without the dataset introduction and description before.
  6. Please add more details on the data preprocessing and augmentation techniques utilization in the section with dataset description.
  7. GAN models are already often used on various tasks. Please compare performance of proposed method and some state-of-the-art for GAN models, used in the similar tasks.
  8. Figure 8 and 9 with more detailed analysis and description of features are not informative. Try to make the interpretation of obtained features in terms of remote sensing.
  9. Figure 11 should be located in the section 3.9.
  10. There is a lack of discussion of other possible ways for the proposed approach utilization or adaptation to solve other remote sensing problems.

Reviewer 3 Report

A very well-written paper with a promise to take an adversarial attack which could be a burning issue to deflecting important information from satellite images.  I have the following comments

  1. Figure 3 seems to be a bit redundant unless you have added any novelty into the unet architecture
  2. Figure 7 has some artifacts around the borders. I am not sure what it is. could you please elaborate more on that.
  3. Figure 10 and 11, looks nice but very hard to read, could you please improve on the readability of the texts of the figures.
  4. In some of the resulting figures, if you could show with pointer, what is significant areas to look at it would be of great help.
  5. A list of the significant contributions at the end of the introduction could be added for the readers.
  6. A demo code in GitHub for the community could be of great help for the readers to explore your proposed method for their own applications.
  7. How general is your proposed methodology when applied to other types of data? Are there any limitations regarding that?